# Experimental Study on Carbonation of Cement-Based Materials in Underground Engineering

**DOI:** 10.3390/ma15155238

**Published:** 2022-07-29

**Authors:** Jun Zheng, Gang Zeng, Hui Zhou, Guanghua Cai

**Affiliations:** 1China Railway 11th Bureau Group Co., Ltd., Wuhan 430061, China; jzheng_crcc@126.com; 2State Key Laboratory of Geomechanics and Geotechnical Engineering, Institute of Rock and Soil Mechanics, Chinese Academy of Sciences, Wuhan 430071, China; 3School of Civil Engineering and Architecture, Hubei University of Arts and Science, Xiangyang 441053, China; 4Hubei Provincial Engineering Research Center of Slope Habitat Construction Technique Using Cement-Based Materials, China Three Gorges University, Yichang 443002, China; 5School of Civil Engineering, Nanjing Forestry University, Nanjing 210037, China; ghcai@njfu.edu.cn

**Keywords:** diversion tunnel, corrosion, cement-based materials, carbonation, experimental study

## Abstract

The corrosive water environment has a decisive influence on the durability of a diversion tunnel lining. In this paper, the effects of carbonation on cement-based materials in water-immersion and saturated-humidity environments were studied by increasing the CO_2_ concentration. The results show that under conditions of water-immersion and saturated humidity, the color of the non-carbonation region is dark, while the carbonation region is gray, and the color boundary is obvious. However, in an atmospheric environment, there is no zone with a dark color and the color boundary is not obvious. In a saturated-humidity environment, the carbonation depth increases over time and changes greatly, and its value is about 16.71 mm at 200 days. While in a water-immersion environment, the carbonation depth varies little with time and the value is only 2.31 mm. The carbonation depths of cement mortar samples in different environments generally follow a linear relationship with the square root of time. The carbonation coefficient in a saturated-humidity environment is more than nine times that in the water-immersion environment. In a water-immersion environment, the carbonation causes a large loss of calcium in cement-based materials, and their Ca/Si ratio obviously decreases. The calcium silicon ratio (Ca/Si) of cement-based materials in a water-immersion environment is 0.11, which is much less than 1.51 in a water-saturated environment and 1.49 in an atmospheric environment. In a saturated-humidity environment, the carbonation only reduces the pH of the pore solution in the carbonation region, and the structural stability of cement-based materials is not degraded. The number of pores of all radii after carbonation in a water-immersion environment exceeds that in a saturated-humidity environment, and the total pore volume and average pore radius in a water-immersion environment are also larger than in a saturated-humidity environment, so the water-immersion environment accelerates the development and expansion of pores. The research results can provide some theoretical and technical support for the design, construction, and safe operation of diversion tunnel linings.

## 1. Introduction

In recent years, the erosion effect of CO_2_ on concrete structures has received increasing attention [1,2,3]. The increased CO_2_ concentration will intensify the carbonation of concrete structures [4,5]. Carbonation is considered one of the most disruptive factors that can affect concrete durability, potentially causing a significant reduction in the service life [6,7]. The degradation of cement-based materials due to CO_2_ attack mainly includes the following chemical reactions [8].
CO_2(g)_↔CO_2(aq)_(1)
CO_2(g)_ + H_2_O↔H_2_CO_3(aq)_↔H^+^(aq) + HCO_3_^−^_(aq)_↔CO_3_^2−^_(aq)_+2H^ + ^_(aq)_(2)
Ca(OH)_2_(s)↔Ca^2+^(aq)+2OH^−^(aq)(3)
Ca^2+^_(aq)_+HCO_3_^−^_(aq)_+OH^−^_(aq)_→CaCO_3(s)_+H_2_O(4)
Ca^2+^_(aq)_+CO_3_^2−^_(aq)_→CaCO_3(s)_(5)
C-S-H_(s)_+H^+^_(aq)_+HCO_3_^−^_(aq)_→CaCO_3(s)_+SiO_x_OH_x(amorphous)_+H_2_O(6)
C-S-H_(s)_+2H^+^_(aq)_+CO_3_^2−^_(aq)_→CaCO_3(s)_+SiO_x_OH_x(amorphous)_+H_2_O(7)
CaCO_3(s)_+CO_2(aq)_+H_2_O↔Ca^2+^_(aq)_+HCO_3_^−^_(aq)_(8)
H^+^_(aq)_+CaCO_3(s)_↔Ca^2+^_(aq)_+HCO_3_^−^_(aq)_(9)

There are four stages in the carbonation process of cement-based materials. In the first stage, some of the CO_2_ dissolves into water (Equation (1)) and some reacts with H_2_O to form carbonic acid (Equation (2)) in the presence of water or humidity. In the second stage, the carbonic acid reacts with hydration products such as Ca(OH)_2_ and C-S-H (Equations (3)–(7)). Carbonation in cement-based materials does not occur at the same time in each phase. When Ca(OH)_2_ is consumed, C-S-H is decomposed into CaCO_3_ and silica gel, reducing the strength of cement-based materials [9]. As the Ca(OH)_2_ is reduced, the pH value of the pore solution inside the cement-based materials falls below about 9.5, which will cause the corrosion of embedded steel in the cement-based materials. In Equations (6) and (7), SiO_x_OH_x_ is an amorphous silica gel [10,11], and the C-S-H gel is converted to a low-C/S gel because of the decalcification, causing the conversion of a portion of the high-density (HD) C-S-H gel to a low-density (LD) C-S-H gel [12,13]. In the third stage, the resulting insoluble CaCO_3_ is transformed into a soluble phase, and this process eventually causes the calcium in the major cement phases to dissolve out of the cement, leaving only a porous silica gel [10]. The fourth stage is referred to as full carbonation.

Carbonation is a diffusion process involving the transportation of CO_2_ from one area to another by way of random molecular motion. Under steady-state conditions, the diffusion process follows Fick’s first law, set out in Equation (10) [14]:X = K_acc._t^n^,(10)
where X is the carbonation depth (mm), K_acc._ is the carbonation coefficient (mm/√year), t is the time duration of exposure (years), and n is an exponent that is smaller than 1.0 (usually taken as 0.5).

At present, various laboratory experiments and models have been developed to understand the erosion of CO_2_ on cement-based materials. Mohammed et al. [15] used 100% CO_2_ concentration to achieve an accelerated carbonation test and studied the carbonation depth and carbonation products. In the research above, the carbonation test had been achieved with a certain humidity of CO_2_, called “dry carbonation” [10]. In this case, the carbonation process generally stops after the creation of CaCO_3_ and acts to increase the strength of the cement and decrease its permeability [16,17,18]. But in some cases, cement-based materials are in a water-immersion environment such as a diversion tunnel. In contrast, “wet carbonation” occurs on cement-based materials that are submerged in liquid (water or brine), and it is similar to the in situ condition of cement-based materials in diversion tunnels.

## 2. Materials and Methods

### 2.1. Materials

The coarse aggregate in the concrete of a diversion tunnel lining is large compared to the size of samples that are used in carbonation tests. In addition, CO_2_ only has an erosion effect on the cement composition of the concrete. Therefore, cement mortar was used as the cement-based material to study the erosion effect of CO_2_ on concrete.

Yellow sand, passed through a 3 mm sieve, was used as the aggregate of cement mortar. The mineral composition of yellow sand was analyzed by a D8-ADVANCE X-ray diffractometer; the main mineral composition is shown in Table 1. Composite Portland cement P•C 32.5R from Huaxin Cement (Wuhan, China) Co., Ltd. was used to produce the cement mortar.

### 2.2. Mixture Design and Sample Preparation

The cement mortar mixture was prepared with an effective water/cement ratio (W/C) of 0.4, and the mass ratio of the cement to yellow sand was 1:1. For the cement mortar mix above, a series of cement mortar samples were cast in plastic molds with dimensions of 150 × 150 × 150 mm^3^ and were then vibrated on a vibration table. After being cured in the laboratory for 24 h, all the cement mortar samples were demolded and further cured in a saturated solution of Ca(OH)_2_ at a temperature of 20 ± 2 °C. After curing for 28 days, the cylindrical samples with dimensions of Φ50 × 100 mm were prepared by core drilling and cutting machines.

### 2.3. Accelerated Carbonation Test Method

It is well known that carbonation is a very slow process in the natural environment, and the service environment of a diversion tunnel is complex. An accelerated carbonation test is crucial to simulate the erosion of the concrete lining in a diversion tunnel. There were two types of accelerated carbonation environments that were utilized in the present research according to the service environment of a diversion tunnel [19]. The schematic diagram of the accelerated carbonation test is shown in Figure 1; a CO_2_ tank was used to provide the vessel with a 99.99% CO_2_ concentration at a pressure of 2 bars. The samples of group HA were placed on a stainless-steel shelf above the deionized water, and the samples of group WB were in the water. Groups HA and WB simulated a saturated-humidity environment and water-immersion environment, respectively. A third group, AI was the control group, and its samples were in an atmospheric environment. The apparatus of the accelerated carbonation test is shown in Figure 2. The test procedure was as follows:(1)The tops and bottoms of the standard cylindrical samples were sealed with dissolved paraffin to ensure the radial movement of the carbonation.(2)The samples of group WB were placed in the bottom of the vessel and deionized water was injected into the vessel. The water surface was 50 mm from the stainless-steel shelf, and the samples of group HA were placed on the stainless-steel shelf as shown in Figure 1. The samples of group AI were placed in an atmospheric environment.(3)Before CO_2_ gas injection, the vessel was vacuumed to the −1 bar by a vacuum pump. The CO_2_ pressure in the vessel was then controlled by a gas regulator, and kept at a required constant level. Keeping the room temperature constant at 20 ± 5 °C, the samples of group AI were in the same laboratory environment.(4)We checked the value of the pressure gauge and the temperature of the laboratory regularly, and changed the water every five days.

### 2.4. Measurement of Carbonation Depth

Cylindrical cement mortar samples were prepared for carbonation depth measurement. At ages of 6, 36, 72, 100, and 200 days, the samples were removed from the vessel and dried in an oven at 60 °C for 48 h. The dried samples were then cut through the middle with a cutting machine. The section surface was cleaned with a brush and the carbonation depth was detected by spraying 1% phenolphthalein-alcohol solution. After 30 s, phenolphthalein pink disappeared in the full-carbonation region that indicated a drop of the pH value below 9.5, as recommended by the International Union of Laboratories and Experts in Construction Materials, Systems, and Structures (RILEM). The carbonation depth was measured with a vernier caliper, and the average carbonation depth was taken as the carbonation depth of the samples.

### 2.5. Phase Changing Monitoring

The phase changing of the carbonation layers of cement mortar samples in different groups was detected by X-ray diffraction (XRD) and thermo gravimetric-differential thermal analysis (TG-DTA). The powder samples were obtained from the carbonation layers by a grinding machine, and were passed through a 75 μm sieve, which was used in XRD and TG-DTA to detect the carbonation products in different groups. The powder samples were first subjected to XRD analysis. They were analyzed by means of a D8-ADVANCE (German) X-ray diffracto meter. The same powder samples that were used in XRD analysis were used in the TG-DTA analysis using a DTG-60 thermo gravimetric-differential thermal analyzer. The DTG-60 series from Shimadzu performed in the range from ambient to 1100 °C with an increase rate of 10 °C/min using a parallel guide differential top pan balance mechanism that simultaneously measured the temperature changes and mass changes between an inert reference and the sample.

### 2.6. Energy Dispersive Spectroscopy (EDS)

The carbonation layers of the samples were cut into small blocks with a particle size of 3–5 mm; a relatively flat surface in the small blocks was required for the test. Before the experiment, the flat surface had to be sprayed with a 10 nm layer of gold-palladium to improve the conductivity. The microstructure of the samples was observed with an FEI QUANTA-250 environmental scanning electron microscope (SEM) that was operated at an accelerating voltage of 30 kV. Energy-dispersive spectroscopy (EDS) was applied to detect the distribution of specific elements such as Ca in the samples.

### 2.7. Quantitative Analysis of the Pore Structure

Low-pressure N_2_ adsorption/desorption experiments were completed with a Quantachrome NOVA1000e series surface area analyzer for surface area and pore analysis. The carbonation layers of the samples were crushed to pass through a 3 mm sieve, and then were dried at 105 °C for 24 h in a vacuum oven. The surface area was calculated from the N_2_ adsorption data in the relative pressure (P/P_0_) range of 0.05–0.35 using the Brunauer-Emmete-Teller (BET) method. The pore volume and pore radius distribution were obtained from the N_2_ adsorption data in the relative pressure (P/P_0_) range of 0.06–0.99 by applying the Barrette–Joynere–Halenda (BJH) method.

## 3. Results

### 3.1. Visual Inspection and Carbonation Depth

Figure 3 presents an exposed sample in the group WB; there was an unknown material precipitation on the surface of the sample. The aggregate of the sample surface was exposed. The unknown precipitation of the sample surface was rinsed with deionized water, and the rinsed liquid was precipitated as shown in Figure 3b. According to a visual inspection, the precipitation contained yellow sand from the cement mortar. Due to the erosion of the saturated carbon dioxide aqueous solution, the cement composition on the surface of the sample was damaged, and the sand fell from its surface. The precipitated solution was oven-dried as shown in Figure 3c.

Under the influence of the infiltration of water pressure and current scour, the effect of CO_2_ on the degradation of concrete was exacerbated. Photos of the degraded diversion tunnel are shown in Figure 4. Figure 4a shows the diversion tunnel of the Lanzhou water source project under construction; the presence of CO_2_ in the groundwater caused the surface of the diversion tunnel lining to precipitate white matter under the action of groundwater seepage pressure. Figure 4b shows the diversion tunnel of the Guangzhou water storage power plant in operation; the aggregate on the concrete surface was exposed under the action of degradation and current scour. The degradation phenomena that are shown in Figure 3 and Figure 4 are very similar.

Figure 5a shows cross-sectional drawings of the samples in different exposed environments (different groups) after cutting. The carbonation layers appear in the marginal region with a white color, which is a typical phenomenon for carbonation. The colors of the internal regions of the exposed samples obviously were quite different. The internal regions of the samples in the atmospheric environment (group AI) were light grey, while those of the samples in the other two environments (groups WB and HA) were dark grey. The color boundary was not obvious in the atmospheric environment but was obvious in the other two environments (groups WB and HA). Figure 5a shows that the color boundary was homogeneous in the water-immersion environment (group WB) and inhomogeneous in the saturated-humidity environment (group HA). The region with a white color in the water-immersion environment (group WB) was smaller than that in the saturated-humidity environment. Figure 5b shows photos and dye testing images of fracture surfaces of the exposed samples after the Brazil splitting test. Half of the fracture surfaces were dropped in the phenolphthalein reagent, while the other half were not dropped. It can be seen from Figure 5b that the white-colored area of a sample is the carbonation area. The sample of group HA shows more uneven visual degradation than the sample of group WB, suggesting a deeper carbonation depth in the former.

Figure 6 shows the carbonation depth with exposure time in different environmental conditions (groups HA and WB). As shown in Figure 6a, the carbonation depth changes little with exposure time in the water-immersion environment (group WB) and is relatively uniform. It can be seen from Figure 6b that the carbonation depth increases with the exposure time in the saturated-humidity environment (group HA), and the sample of the group HA shows uneven carbonation. Comparing Figure 6a,b, we see that carbon dioxide on the carbonation of the cement mortar is more obvious in the saturated-humidity environment (group HA) than in the water-immersion environment (group WB).

Figure 7 shows the change of the carbonation depth of cement mortar with exposure time due to an accelerated carbonation test with 100% CO_2_ for up to 200 days. The samples in the saturated-humidity environment (group HA) show a higher carbonation depth over the entire time than the samples in the water-immersion environment (group WB). The carbonation depth of the samples in the saturated-humidity environment is about 16.71 mm, whereas the carbonation depth of the samples in the water-immersion environment is only 2.31 mm. In Figure 7, the carbonation depths of the cement mortar samples in different environments generally follow a linear relationship with the square root of time. The experimental results show an excellent correlation coefficient with linear regression. The accelerated coefficient of diffusion for the cement mortar samples in different environments was calculated as the slope of the relationship between the carbonation depth and the square root of time, as shown in Figure 7, according to Fick’s first law (Equation (10)). The carbonation coefficient in a saturated-humidity environment is more than nine times that in the water-immersion environment from the linear fitting equation in Figure 7.

### 3.2. X-ray Diffraction Pattern Analysis

X-ray diffraction was used to identify the crystalline phases that were present in the precipitate (Figure 3c). The X-ray diffraction pattern is shown in Figure 8. The most intense peak on the pattern is located at 26.6° (2θ) and corresponds to the crystallographic plane in quartz (SiO_2_). Several other quartz (SiO_2_) peaks are also shown. Except for quartz (SiO_2_) peaks, the most intense peak on the pattern is located at 27.9° (2θ) and corresponds to the crystallographic plane in albite. Less intense peaks corresponding to microcline maximum (KAlSi_3_O_8_), calcium aluminum oxide, calcite (CaCO_3_), and dolomite [CaMg(CO_3_)_2_] that are are also present in this pattern, and are labeled. The strongest calcite (CaCO_3_) peak is located at 29.4° (2θ).

The principle of obtaining the mass percentages of chemical phases is that the intensity of X-ray diffraction is proportional to the mass percentages of the detected phases in the samples. The contribution of chemical phases of the precipitate (Figure 3) is shown in Figure 9. More than half of the precipitate is composed of quartz (SiO_2_). The precipitate also contains albite and microcline maximum, with mass percentages of 22.69% and 11.67%, respectively. It can be seen from Table 1 that the sand contains albite and microcline maximum, while there is no albite or microcline maximum in the cement, so the precipitate contains a certain amount of yellow sand. As shown in Figure 3, the liquid after rinsing the surface of the sample that was exposed to a water-immersion environment contains sand, and the apparent result is consistent with the XRD result. The precipitate contains small amounts of calcium aluminum oxide (9.14%) and calcite (CaCO_3_) (3.25%).

Figure 10 presents the XRD patterns of the carbonation layers of the samples that were exposed to different environments. Peaks are identified in the diffraction patterns relating to quartz (SiO_2_), calcite (CaCO_3_), albite, microcline maximum (KAlSi_3_O_8_), and dolomite [CaMg(CO_3_)_2_] in different groups. Due to carbonation, the calcium hydroxide (Ca(OH)_2_) peaks are of lower intensity or are not visible in the patterns that are shown in Figure 10. The patterns corresponding to the powder of carbonation layers in different environments all show the most intense peak at 26.6° (2θ), and the peak corresponds to the crystallographic plane in quartz (SiO_2_). Except for the quartz (SiO_2_) peaks, the most intense peaks on the patterns are all located at 29.4° (2θ) and correspond to the crystallographic plane in CaCO_3_. It can be observed from Figure 10 that the peak intensity of CaCO_3_ in the atmospheric environment is smaller than in the other two environments.

The contribution of chemical phases of carbonation layers in different environments is shown in Figure 11. As both the cement and sand contain quartz (SiO_2_), the mass percentage of quartz (SiO_2_) is higher than other chemical phases in different exposed environments. As shown in Figure 11, there are some differences in the mass percentages of carbonation product CaCO_3_ in different environments. The mass percentage of CaCO_3_ of the carbonation layer in the saturated-humidity environment is the highest, about 17.34%, followed by the water-immersion environment, about 13.91%, while the mass percentage of CaCO_3_ in an atmospheric environment is the least, about 11.59%.

### 3.3. Thermogravimetric-Differential Thermal Analysis (TG-DTA)

Thermo gravimetric-differential thermal analysis (TG-DTA) is a method for analyzing the chemical composition and content of chemical substances by measuring the change of the mass at different temperatures. As the temperature rises, the bound water in calcium hydroxide (Ca(OH)_2_) is generally decomposed between 300 °C and 500 °C, while CO_2_ in calcium carbonate (CaCO_3_) is decomposed between 600 °C and 700 °C. Due to the above decomposition, calcium hydroxide and calcium carbonate are decomposed into calcium oxide.

Many factors affect the thermal decomposition temperature of Ca(OH)_2_ and CaCO_3_. Since the range of thermal decomposition temperature is not fixed, it should be determined by the endothermic peak of the differential thermal analysis (DTA) curve. The difference in the thermo gravimetric analysis (TGA) curves is calculated from the range of the thermal decomposition temperature, which is the mass percentage of the bound water that is released by the Ca(OH)_2_ decomposition process and the mass percentage of CO_2_ that is released during the CaCO3 decomposition process. Depending on the mass percentage of H_2_O and CO_2_, the mass percentage of Ca(OH)_2_ and CaCO_3_ can be determined. The TG-DTA curves of the carbonation layers in different exposed environments are shown in Figure 12.

In Figure 12, the endothermic peak of thermal decomposition of CaCO_3_ is visible, while the endothermic peak of thermal decomposition of Ca(OH)_2_ is weak. The temperature range of the thermal decomposition of CaCO_3_ was determined according to the endothermic peak, and the content of CaCO_3_ was further determined as shown in Table 2. It can be seen that the percentage of CaCO_3_ in the carbonation layer of the sample in the saturated-humidity environment is about 15.58%, and the content of CaCO_3_ in the carbonation layer of the sample in the water-immersion environment comes second, about 13.91%; meanwhile, that in the atmospheric environment is the least, about 9.47%.

### 3.4. Energy Dispersive Spectroscopy (EDS) Analysis

The energy dispersive spectroscopy (EDS) spectra were used to verify the elemental compositions of crystals on the surfaces of carbonation layers of the samples by an energy dispersive spectrometer using a scanning electron microscope (SEM). Energy dispersive spectroscopy (EDS) analysis can only test the type and content of elements of a tiny region on the surface of a sample. The results of a single point are very random and cannot represent the overall properties of the sample. Therefore, several spectral points were selected. Figure 13 presents the SEM images of the carbonation layers of the samples and the selected spectral points on the SEM images for EDS analysis.

Figure 14 presents the typical EDS spectra that were acquired at spectral points 5, 3, and 3 in Figure 13a–c, respectively. Figure 14a,b show the presence of carbon (C), oxygen (O), magnesium (Mg), aluminum (Al), silicon (Si), and calcium (Ca), which are the main elements of spectra 5 and 3 in Figure 13a,b, respectively. Figure 14c shows the presence of carbon (C), oxygen (O), silicon (Si), and calcium (Ca), which are the main elements of spectrum 3 in Figure 13c. In Figure 14, the EDS spectra clearly show that the intensity of the Ca peak in the water-immersion environment is weaker than in the other two environments.

Table 3, Table 4 and Table 5 show the element composition that was obtained from the EDS spectra, expressed as the weight percentage (wt%) of the selected spectral points at the carbonation layers of samples in different environments. Since these selected spectral points are limited and the distribution of chemical compounds is rather heterogeneous in these materials, these results should be considered only relative and not absolute. According to these results, the average mass percentage content (wt%) of calcium in the saturated-humidity environment is about 18.55%, it is about 19.43% in the atmospheric environment, and each selected spectral point in the two environments contains the Ca element. Not all the selected spectral points contain Ca in the water-immersion environment, and the average mass percentage content (wt%) of Ca is only 3.46%, which is much lower than in the other two environments. The calcium-silicon ratio (Ca/Si) represents the mass ratio of calcium and silicon; the calcium-silicon ratios (Ca/Si) are about 1.49 and 1.51 in the atmospheric and saturated-humidity environments, respectively. Due to the calcium leaching process of the cement-based materials in the water-immersion environment, the calcium-silicon ratio (Ca/Si) is much less than in the other two environments, with a value of approximately 0.11.

### 3.5. Quantitative Analysis of the Pore Structure

In general, carbonation can improve the pore structure of the matrix for cement-based materials. However, the impact of carbonation on the pore structure varies in the different exposed environments, as shown in Figure 15, which represents the pore radius distribution from the adsorption branch of isotherms using the BJH method. It is observed that the pore radius in the pore structure of carbonation layers in the different groups was between 1.6 and 50.4 nm, and all presented a major peak at around 13 nm. The pore radius distribution curve of the carbonation layers of samples in the water-immersion environment (group WB) is above that in the saturated-humidity environment (group HA). The pore radius distribution curves of groups AI and HA intersect at a pore radius of 11 nm, while the pore radius distribution curves of Group AI and WB intersect at a pore radius of 30 nm. For a pore radius that is less than 11 nm, the lower values of pore volume are in group AI, while the group HA has higher values of pore volume. However, we see the opposite result for a pore radius that is greater than 11 nm. In most of the pore radius ranges (less than 30 nm), the pore volume of group WB is larger than that of group AI, and only in the range of pore radii that are greater than 30 nm is the pore volume of group AI larger.

The pore structure parameters of the carbonation layers in different environments that were determined by the nitrogen adsorption method are shown in Table 6. The specific surface area that was calculated from N_2_ adsorption data using the multipoint BET model ranges from 23.6899 to 35.3504 m^2^/g, and the specific surface area of group HA is the largest, at about 35.3504 m^2^/g, while that of group AI is the smallest, at about 23.6899 m^2^/g. The specific surface area of group HA is 1.5 times that of group AI, and it is only 3.058 m^2^/g larger than that of group WB. The total pore volumes (cm^3^/g) were found to be 0.0671 for group AI, 0.0787 for group WB, and 0.0681 for group HA (Table 6). The average pore radius of group AI, 5.6623 nm, is larger than those of groups WB and HA, and the average pore radius of group HA is the smallest of all groups.

## 4. Discussion

Carbonation is a long-term durability problem of reinforced concrete, predominantly for two reasons. It causes the pH value of a pore solution inside cement-based materials to fall, which leads to corrosion of the steel bar in the cement-based materials, and it also causes degradation of the cement-based materials. As shown in Figure 3, the cement composition on the surface of the sample is damaged, and hence continuous degradation of the surface results in exposure of the sand in the water-immersion environment. Initially, CO_2_ causing the erosion of cement is mainly due to the carbonation of Ca(OH)_2_, and CaCO_3_ is formed, which fills the pores of the cement-based materials, reducing the porosity and permeability but increasing strength. As erosion continues, Ca(OH)_2_ is consumed, and C-S-H gel is decomposed into CaCO_3_ (Equation (7)). In the water-immersion environment, CaCO_3_ is converted to water-soluble Ca(HCO_3_)_2_, which will increase the porosity and permeability but result in the loss of strength. As shown in Figure 3, the sand on the surface is easily rinsed off with deionized water. The presence of water causes CO_2_ to have a severe erosion effect on the cement of cement mortar, destroying the chemical stability of cement hydration products, resulting in the decrease in the cementation capacity of cement. Much research has shown that a water environment greatly influences the corrosion process of cement-based materials [9]. However, the carbonation depth in a saturated-humidity environment is much deeper than in a water-immersion environment (Figure 5 and Figure 6). As shown in Figure 7, the carbonation depth and square-root time follow a linear relationship in the water immersion and saturated-humidity environments, but the carbonation rate in the saturated-humidity environment is more than nine times that in the water-immersion environment. El-Turki et al. [20] proposed that the rate of carbonation rate increases with the relative humidity. High humidity provides the most favorable conditions for carbonation (Equation (2)), and it improves the carbonation rate. However, the carbonation rate dropped sharply in the water-immersion environment. The carbonation process is ongoing due to the diffusion of CO_2_ in the pore network according to Fick’s first law, and the diffusion is much faster in air than in water. The presence of water decreased the carbonation rate due to the slow diffusion of CO_2_ within the saturated pore network, whereas the excessive water could cause deterioration to the cement-based materials in the longer term. Figure 5 presents color-coded images before dropping the phenolphthalein reagent, and color changes, through visual inspection, after dropping the phenolphthalein reagent. These pictures show the interface between the carbonation and non-carbonation zones. Kutchko et al. [8] pointed out that un-hydrated cement compounds are usually the brightest, while calcium hydroxide (CH) is less bright, and calcium silicate hydrate(C-S-H) is darker still. Water is the most important factor affecting the hydration degree of cement components. Since the moisture content is higher in the water-immersion and saturated-humidity environments, the degree of hydration is greater in these groups. As a result, the color of the middle zone of samples in the water-immersion and saturated-humidity environments is darker than in the atmospheric environment.

According to the XRD and TG-DTA analysis results, the CaCO_3_ increased in all the groups. The content of CaCO_3_ of the carbonation layers in the saturated-humidity and water-immersion environments is higher than in the atmospheric environment, and it is highest in the saturated-humidity environment. As shown in Equations (1)–(5), the content of the carbonation product CaCO_3_ is controlled by several parameters, such as the water content and CO_2_ concentration. The high concentration of CO_2_ and large amounts of water provide favorable conditions for carbonation in the water-immersion and saturated-humidity environments, so the content of CaCO_3_ is high. Water also affects the degree and speed of cement hydration, so the content of hydration products such as Ca(OH)_2_ in the saturated-humidity and water-immersion environments is higher than in the atmospheric environment, which is also why the content of carbonation product CaCO_3_ is lowest in the atmospheric environment. In the water-immersion environment, CaCO_3_ is converted to water-soluble Ca(HCO_3_)_2_ (Equations (1)–(5)), and then the Ca(HCO_3_)_2_ is dissolved in water. Since the stability of Ca(HCO_3_)_2_ is poor, part of Ca(HCO_3_)_2_ is decomposed into CaCO_3_ and covers the surface of the sample. As shown in Figure 9, the precipitate on the surface of the sample in the water-immersion environment contains 3.25% CaCO_3_. However, in the saturated-humidity environment, CaCO_3_ fills the pores of the cement-based materials and does not precipitate from the pores. Therefore, the content of CaCO_3_ of the carbonation layer in the saturated-humidity environment is higher than in the water-immersion environment.

The carbonation layer of the sample had relatively less Ca content in the water-immersion environment, while the Ca content of the carbonation layers in the saturated-humidity and atmospheric environment was higher, and the values in these two environments were almost the same. In the saturated-humidity environment, the erosion effect of CO_2_ on the cement-based materials is mainly the rapid increase of the carbonation depth, which leads to the corrosion of the embedded steel, while in the water-immersion environment, the leaching process of cement-based materials is the main cause of degradation of cement-based materials. Both Ca and Si in cement-based materials are dissolved in the water-immersion environment, but compared with Ca, Si occupies only a small portion, and its dissolution rate is extremely slow [21]. Thus, the silicon (Si) leaching process is ignored, and the calcium-silicon ratio (Ca/Si) can be used to characterize the calcium leaching process of cement-based materials in the water-immersion environment. The average Ca/Si ratio in the water-immersion environment is only 0.14, which is much smaller than the average Ca/Si ratio, which is about 1.5 in the other two environments. Dauzères et al. [22] proposed that if the Ca/Si ratio decreases, the chemical stability of cement hydrated products is destroyed. As shown in Figure 6, the carbonation depth of the sample in the water-immersion environment is much shallower than in the saturated-humidity environment. The decalcification in the water-immersion environment leads to damage of the carbonation layer, while the carbonation layer in the saturated-humidity environment is not damaged, exhibiting only a decrease of the pH value.

Figure 15 shows that the pore volume of small pores in the water-immersion and saturated-humidity environments exceeds that in the atmospheric environment. As shown in Table 6, the specific surface area and total pore volume in the water-immersion and saturated-humidity environments exceed those in the atmospheric environment, while the average pore radius is less than in the atmospheric environment. This indicates that the number of small pores in the water-immersion and saturated-humidity environments is more than in the atmospheric environment. This is mainly due to the crystallization and precipitation of CaCO_3_ in the pores, where large pores are divided into small pores. As carbonation continues to consume the Ca(OH)_2_ of the pore solution, the Ca(OH)_2_ crystal of the cement hydration products is dissolved to supplement the loss of Ca^2+^ in the pore solution, resulting in an increase in the number of small pores in the cement-based materials. Compared with the atmospheric environment, water is abundant in the environments of water-immersion and saturated-humidity, and water is the basic condition for cement hydration. Therefore, the content of Ca(OH)_2_ and C-S-H gels in the later stage are improved. At the same time, water accelerates the rate at which Ca(OH)_2_ crystal dissolves into the pore solution and provides sufficient moisture that is required for the carbonation reaction, and thus the degree of carbonation is more adequate. With the continuous consumption of Ca(OH)_2_ crystals, C-S-H gel will continue to be carbonized by CO_2_, and the high-density C-S-H gel will be converted to low-density C-S-H gel after decalcification of the C-S-H gel. The pore volume of all the pore radii in the water-immersion environment is larger (Figure 15) than in the saturated-humidity environment, and the total pore volume and average pore size in the water-immersion environment exceeds those in the saturated-humidity environment (Table 6). This is mainly because CO_2_ further reacts with the carbonation product CaCO_3_ to form soluble Ca(HCO_3_)_2_, and then precipitates out in the water-immersion environment, leading to the loss of Ca in the cement-based materials. Thus, the water-immersion environment will aggravate the further development and expansion of the pores.

## 5. Conclusions

Diversion tunnels belong to underground structures, the service environments of which are more complicated than concrete structures on the ground. During the operation period, the concrete lining of the diversion tunnel is in a water-immersion environment, while during the maintenance period, it is in a saturated-humidity environment. In this paper, we studied the effect of carbonation on cement-based materials in water-immersion and saturated-humidity environments by using an accelerated CO_2_ erosion test instrument that simulates the service environment of a diversion tunnel. Based on the experimental work in the present study, the following concluding remarks can be made:

(1)In the saturated-humidity environment, the carbonation depth increases with exposure time, which changes greatly, while in the water-immersion environment, the carbonation depth changes little with exposure time. However, the water-immersion environment will cause degradation of the cement-based materials.(2)The color of the non-carbonation region is dark, and that of the carbonation region is gray in the water-immersion and saturated-humidity environments, while there is no dark-colored region in the atmospheric environment. This is mainly due to adequate moisture, which promotes the continued hydration of cement; thus, in the saturated-humidity and water-immersion environments, the cement has a greater degree of hydration.(3)The content of the carbonation product CaCO_3_ is higher in the water-immersion and saturated-humidity environments than in the atmospheric environment. In the water-immersion environment, CO_2_ continues to react with the carbonation product CaCO_3_ to produce soluble Ca (HCO_3_)_2_, which results in a lower content of CaCO_3_ in the water immersion environment than in the saturated-humidity environment. Therefore, in the water-immersion environment, carbonation will cause decalcification of the cement-based materials, which leads to a significant decrease of the Ca/Si ratio in these materials, ultimately destroying the structural stability of cement-based materials.(4)In the water-immersion and saturated-humidity environments, the number of small pores in the cement-based materials is greater than in the atmospheric environment. The number of pores of all the pore radii in the water-immersion environment exceeds that in the saturated-humidity environment, and the pore volume and average pore size are also greater than in the saturated-humidity environment. Thus, the water-immersion environment will aggravate the further development and expansion of the pores, resulting in the increased permeability and reduced strength of cement-based materials.

## Figures and Tables

**Figure 1 materials-15-05238-f001:**
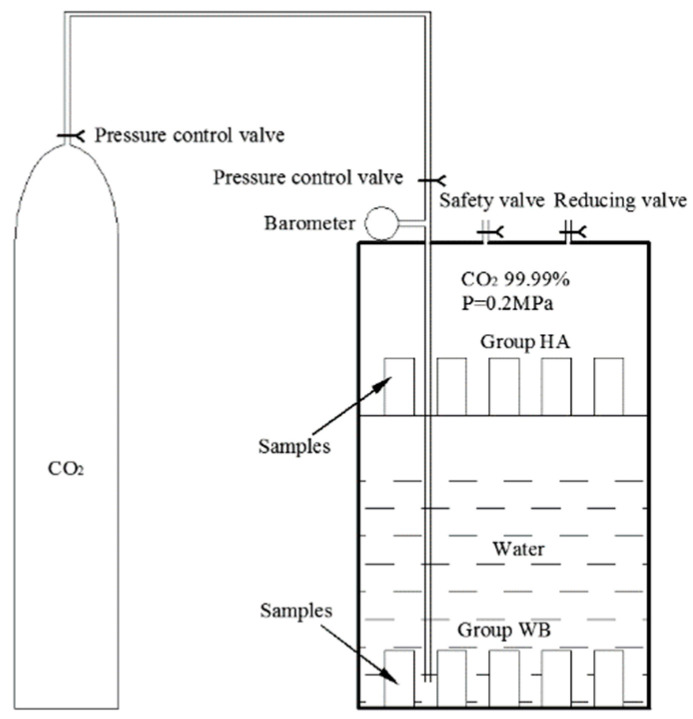
Schematic diagram of the pressurized accelerated carbonation test equipment.

**Figure 2 materials-15-05238-f002:**
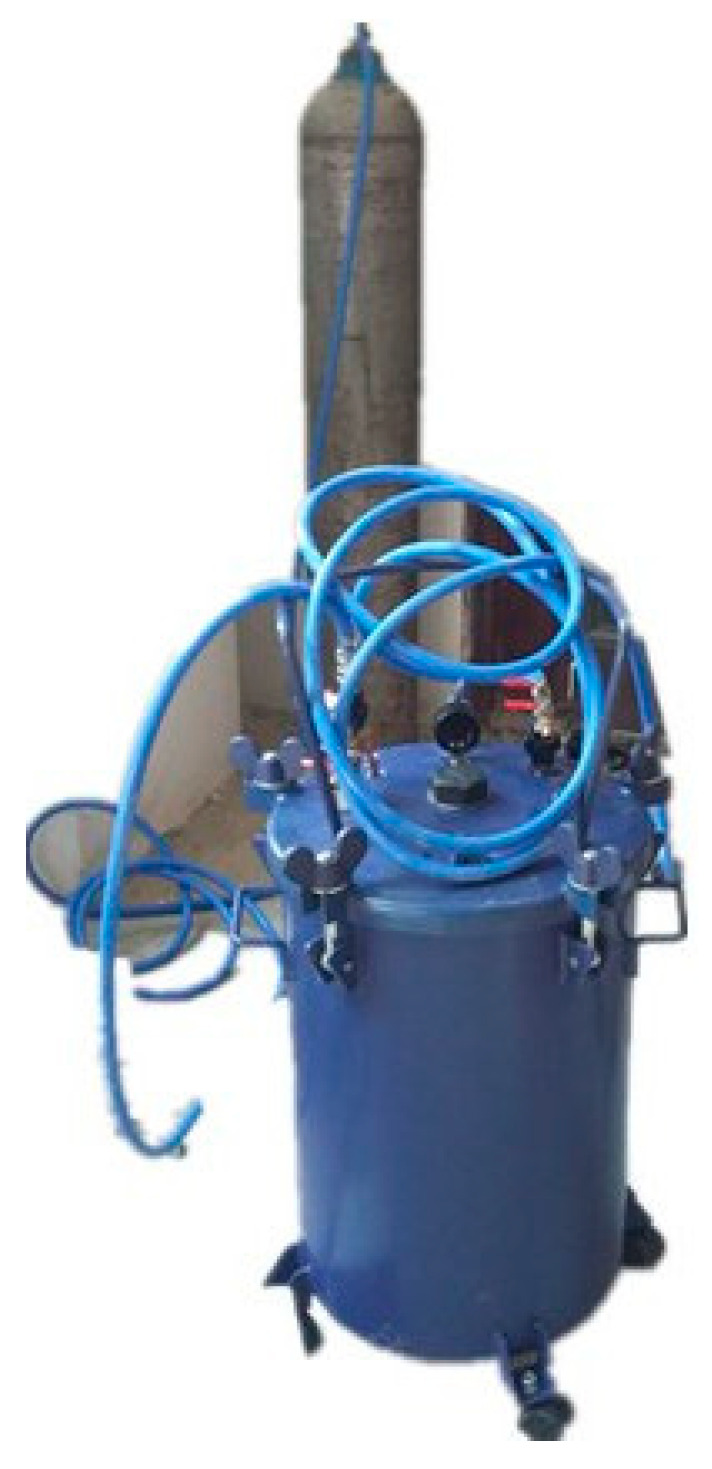
Apparatus of the accelerated carbonation test.

**Figure 3 materials-15-05238-f003:**
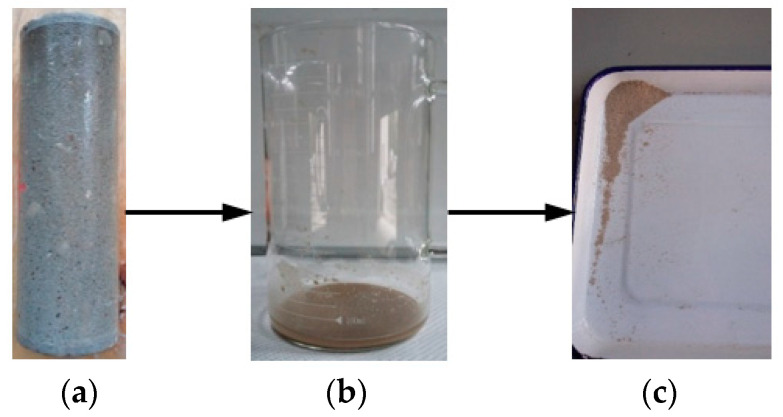
Photos of the exposed sample in group WB (water-immersion environment); (**a**) sample, (**b**) precipitate, and (**c**) oven dry.

**Figure 4 materials-15-05238-f004:**
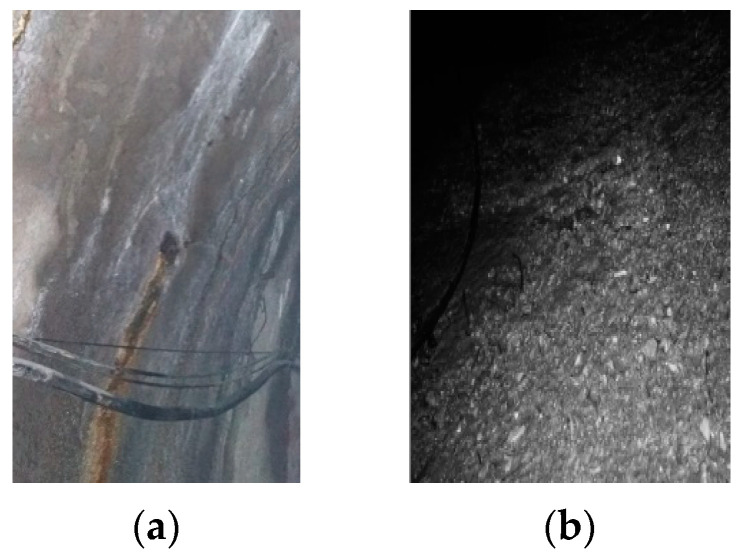
Photos of the degraded diversion tunnels: (**a**) the diversion tunnel of Lanzhou water source project, and (**b**) the diversion tunnel of Guangzhou energy storage power plant.

**Figure 5 materials-15-05238-f005:**
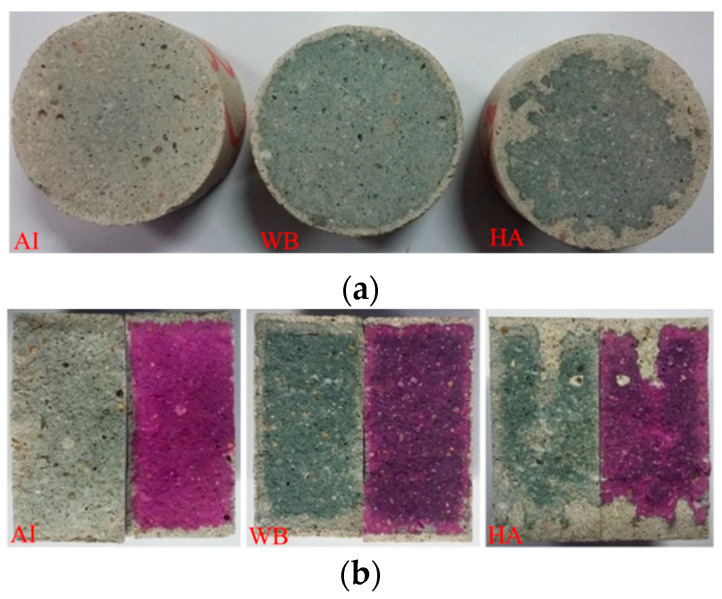
Sectional drawings of the eroded samples: (**a**) cross-section and (**b**) vertical section.

**Figure 6 materials-15-05238-f006:**
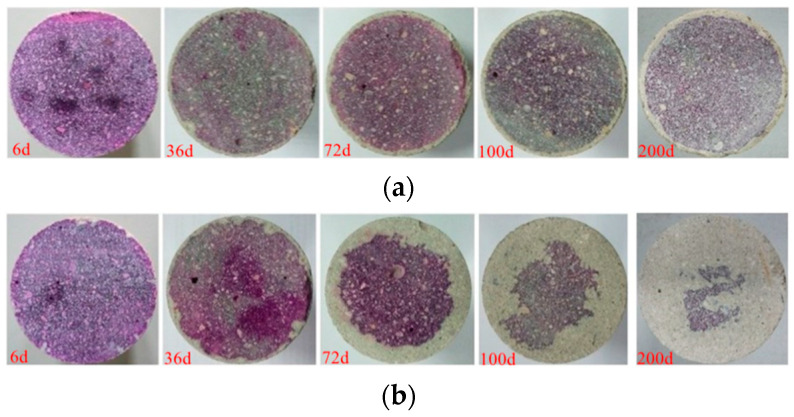
Carbonation depth measurements: (**a**) Group WB (water-immersion environment) and (**b**) Group HA (saturated-humidity environment).

**Figure 7 materials-15-05238-f007:**
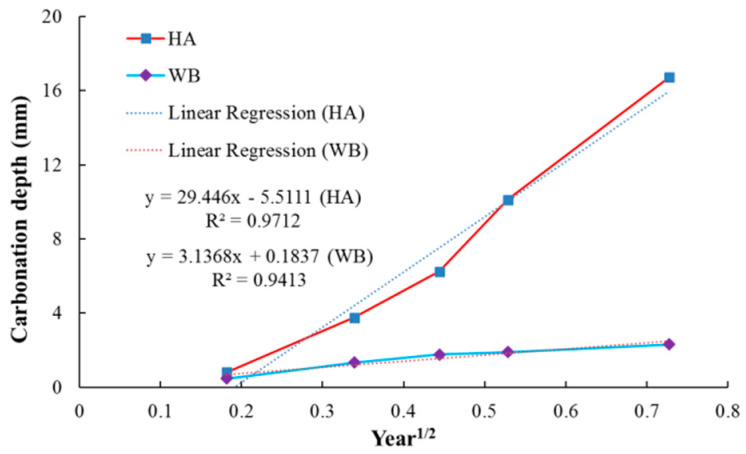
Average carbonation depth versus the square root of exposure time (year) relationships.

**Figure 8 materials-15-05238-f008:**
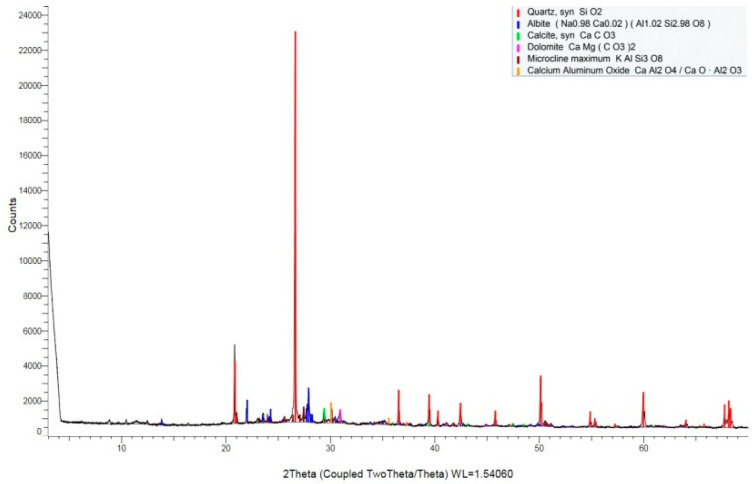
X-ray diffraction pattern of the precipitate.

**Figure 9 materials-15-05238-f009:**
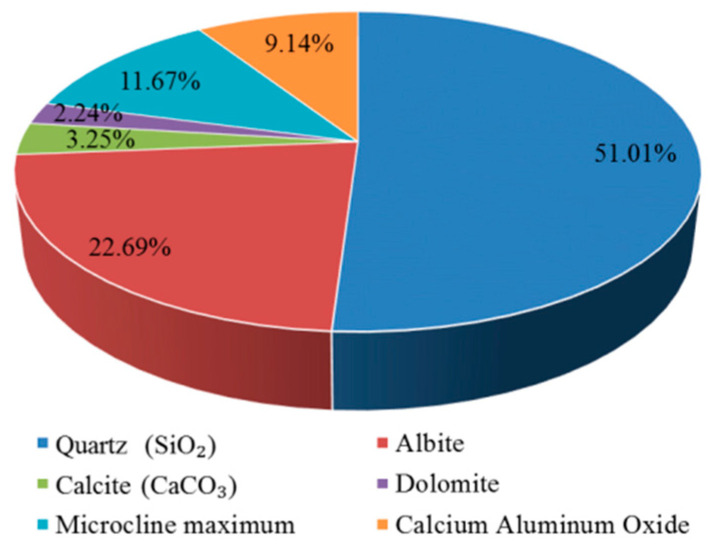
Contribution of chemical phases of the precipitate.

**Figure 10 materials-15-05238-f010:**
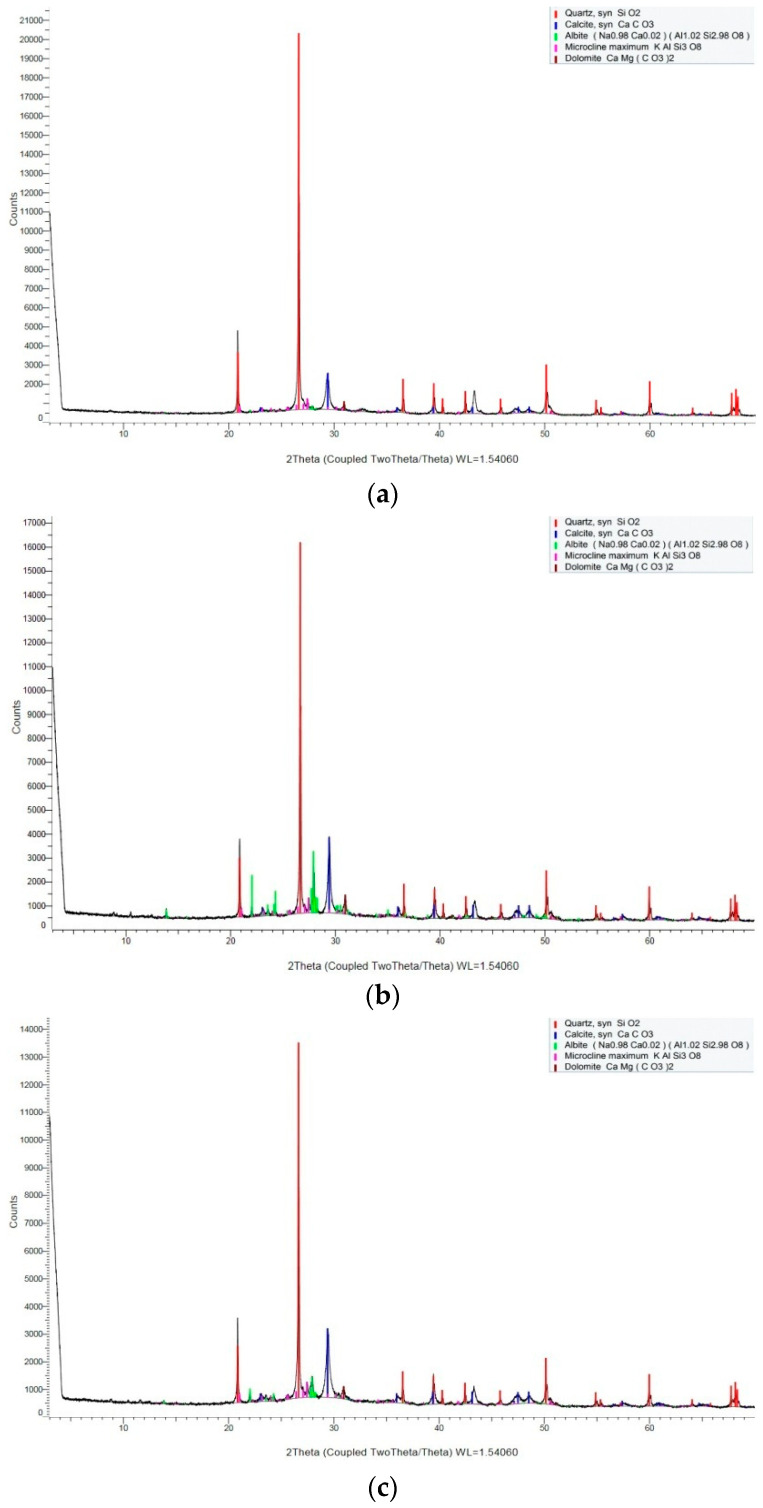
X-ray diffraction patterns of carbonation layers in different environments: (**a**) group AI (atmospheric environment), (**b**) group WB (water-immersion environment), and (**c**) group HA (saturated-humidity environment).

**Figure 11 materials-15-05238-f011:**
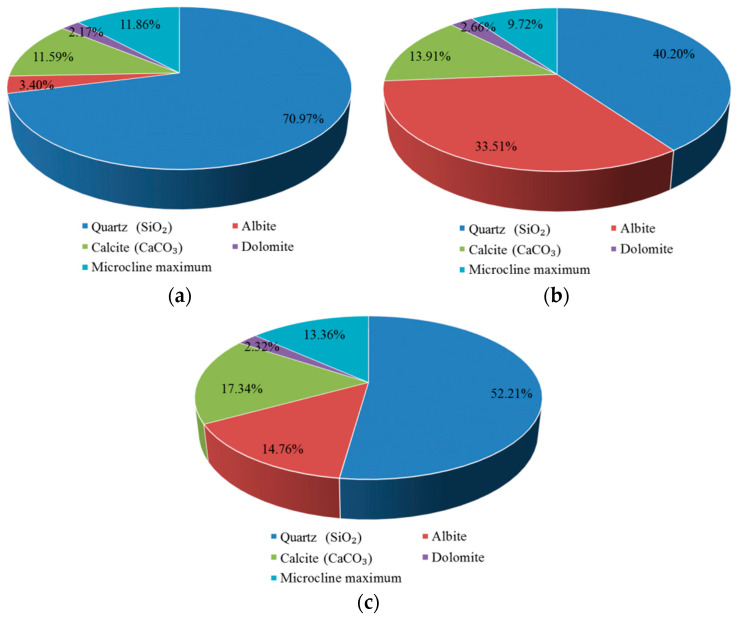
Contribution of chemical phases of carbonation layers in different exposed environments: (**a**) group AI (atmospheric environment), (**b**) group WB (water-immersion environment), and (**c**) group HA (saturated-humidity environment).

**Figure 12 materials-15-05238-f012:**
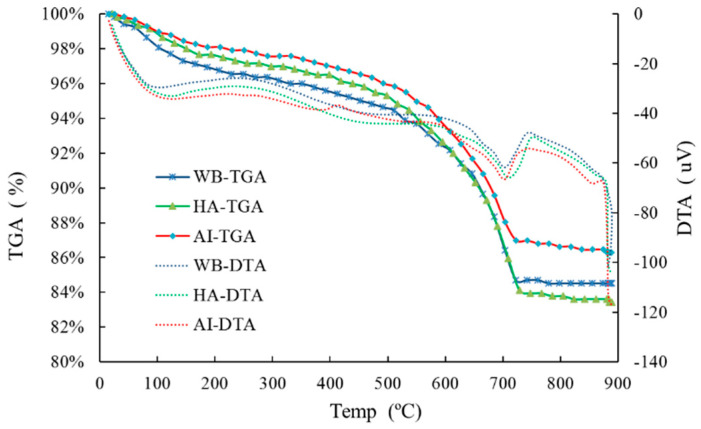
TG-DTA curves of the carbonation layers in different exposed environments.

**Figure 13 materials-15-05238-f013:**
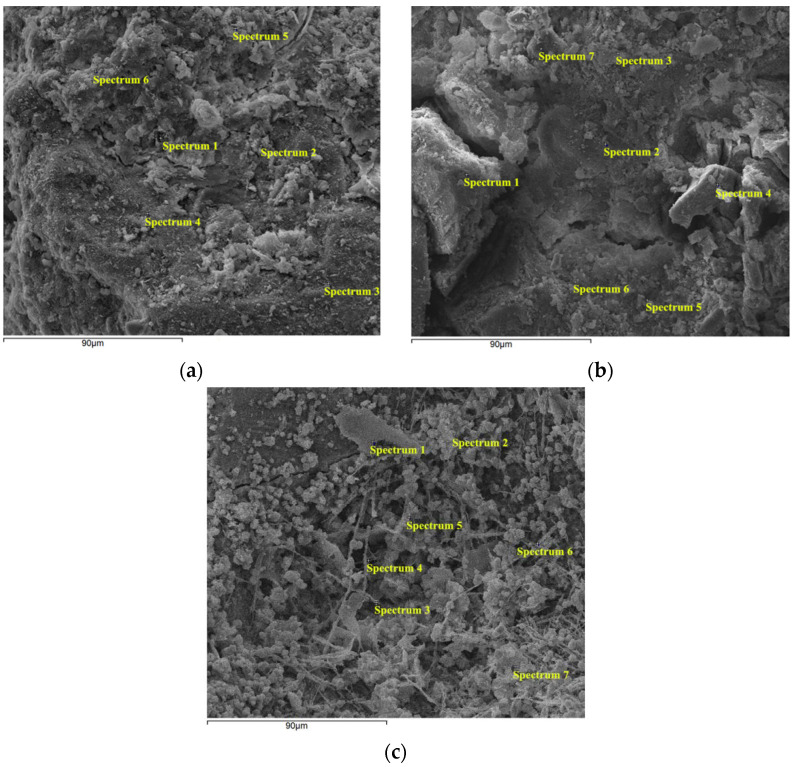
SEM of the carbonation layers of the samples with the selected spectral points for EDS analysis: (**a**) group AI (atmospheric environment), (**b**) group WB (water-immersion environment), and (**c**) group HA (saturated-humidity environment).

**Figure 14 materials-15-05238-f014:**
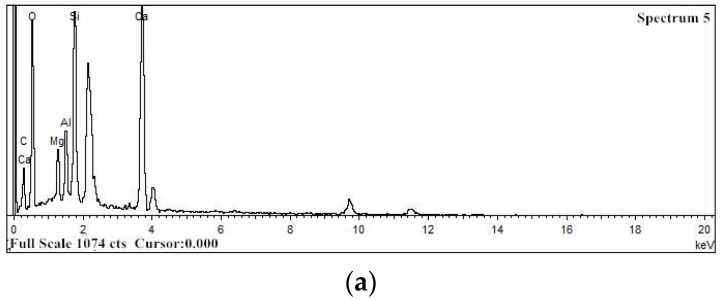
Typical EDS spectra for the marked points in Figure 13: (**a**) group AI (atmospheric environment), (**b**) group WB (water-immersion environment), and (**c**) group HA (saturated-humidity environment).

**Figure 15 materials-15-05238-f015:**
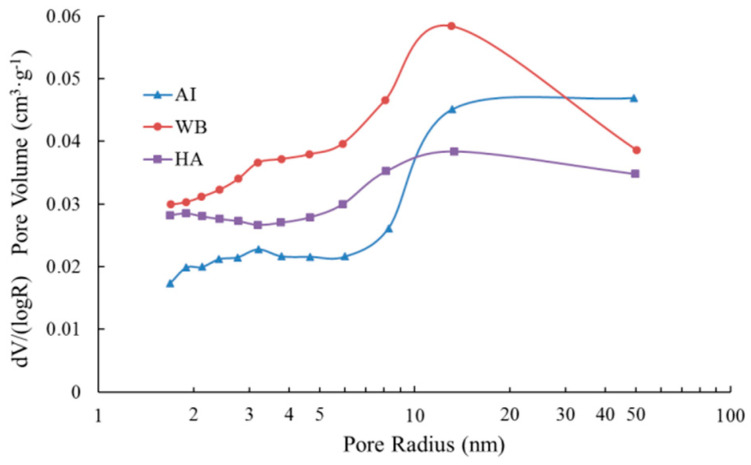
Pore radius distribution of the carbonation layers in different exposed environments.

**Table 1 materials-15-05238-t001:** Main mineral composition of sand.

Mineral Composition	SiO_2_	Albite (Na_0.98_Ca_0.02_) (Al_1.02_Si_2.98_O_8_)	Microcline Maximum (KAlSi_3_O_8_)
Mass Fraction/%	74.66	7.27	18.07

**Table 2 materials-15-05238-t002:** The mass percentage content of calcium carbonate in the carbonation layer of the samples.

Group	T_1_ (°C)	T_2_ (°C)	CaCO_3_ (%)
AI	658.91	736.58	9.47
WB	645.69	741.74	13.91
HA	640.96	747.23	15.58

**Table 3 materials-15-05238-t003:** Element mass percentage content of the carbonation layer in group AI (wt%).

Spectrum	C	O	Mg	Al	Si	Ca	Fe
1	8.58	54.54	1.25	3.90	6.57	25.16	
2	12.28	47.23	3.17	4.83	12.61	19.29	0.59
3	8.72	47.82	0.40	6.62	21.72	14.72	
4	15.19	50.13		5.81	18.30	10.57	
5	15.07	51.83	2.80	3.54	9.42	17.34	
6	8.81	45.30		6.98	9.41	29.50	
Max	15.19	54.54	3.17	6.98	21.72	29.50	0.59
Min	8.58	45.30	0.00	3.54	6.57	10.57	0.00
Average	11.44	49.48	1.27	5.28	13.01	19.43	0.10

**Table 4 materials-15-05238-t004:** Element mass percentage content of the carbonation layer in group WB (wt%).

Spectrum	C	O	Mg	Al	Si	Ca	Fe
1		54.56			45.44		
2	6.83	49.94	4.56	10.95	20.24	4.16	3.32
3	17.73	56.98	4.77	7.5	11.19	1.83	
4	9.44	39.32		11.19	32.91	7.14	
5	10.01	40.06	3.48	11.42	27.41	7.62	
6	5.64	56.82		9.48	24.57	3.49	
7		45.4			54.6		
Max	17.73	56.98	4.77	11.42	54.6	7.62	3.32
Min	0	39.32	0	0	11.19	0	0
Average	7.09	49.01	1.83	7.22	30.91	3.46	0.47

**Table 5 materials-15-05238-t005:** Element mass percentage content of the carbonation layer in group HA (wt%).

Spectrum	C	O	Mg	Al	Si	Ca	Fe
1	9.51	44.34	5.65	6.74	18.77	8.58	6.41
2	19.08	46.12			15.56	19.24	
3	21.96	57.42			9.75	10.87	
4	10.51	37.28		4.63	22.53	25.05	
5	15.08	62.5		3.5	9.71	9.21	
6	13.41	52.44			4.66	29.49	
7	10.13	53.38		3.85	5.24	27.4	
Max	21.96	62.50	5.65	6.74	22.53	29.49	6.41
Min	9.51	37.28	0	0	4.66	8.58	0
Average	14.24	50.50	0.81	2.67	12.32	18.55	0.92

**Table 6 materials-15-05238-t006:** Pore structure parameters of the carbonation layers of the samples.

Group	Specific Surface Area (m^2^·g^−1^) ^a^	Total Pore Volume (cm^3^·g^−1^) ^b^	Relative Pressure (P/P_0_) ^c^	Average Pore Radius (nm) ^d^
AI	23.6899	0.0671	0.9882	5.6623
WB	32.2924	0.0787	0.9884	4.8736
HA	35.3504	0.0681	0.9883	3.8546

^a^ Specific surface area is calculated by the BET method. ^b^ Total pore volume is estimated to be the liquid volume of nitrogen at the relative pressure in Table 6. ^c^ Relative pressure is the pressure that is used to calculate the pore volume. ^d^ Average pore radius = 2 ∗ total pore volume/specific surface area.

## Data Availability

The data that support the findings of this study are available on request from the corresponding authors.

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
