# Peer review of "Experimental Study on Carbonation of Cement-Based Materials in Underground Engineering"

_materials, 2022, doi:10.3390/ma15155238_

Round 1

Reviewer 1 Report

Having now evaluated the manuscript titled "Experimental Study on Carbonation of Cement-based Materials under Water-Immersion and Saturated-Humidity Environment" by Zheng  et all., while I do not question the scientific quality of the work, I do not feel that the perceived conceptual novelty of the manuscript warrants interest to the broad audience of Materials journal.

 1. Does the title, seen in isolation, give a full yet concise and specific indication of the work reported? The authors should pay attention to assuring that the title of their study is clear and informative, and should reflect the aim and approach of the work. Moreover, at a simple search, It was found the title looks similar to other already published papers. So please rewrite it with the respect to the above-mentioned characteristics.  

2. The Abstract of the present study is poorly written and lacks important information, it is, therefore, the duty of the author to ensure that the abstract is properly representative of the entire paper giving more data gathered out during the experiments.

3. The quality of figures and graphs are of very poor quality. The authors should consider that graphs, figures, and tables can save readers time and energy, aid their understanding of a scientific article, and reduce the word count of the main text. All graphs and figures as presented in the present manuscript fail to meet their potential and include mistakes that jeopardize their clarity. When the appropriate format is used to depict data, it conveys the greatest amount of information in the clearest fashion, complements the text, and deepens readers' understanding.

On the other hand, I found some of the descriptions of some very important points were inadequate or completely missing. I have very little confidence in one important analysis such X-ray diffraction, and came away with too many questions to be able to recommend this paper for publication as it stands.. I would recommend to the authors to present the result as superimposed patterns of the samples. As an exemplification, the authors provide the X-ray diffractometry results (Fig 8 and Fig 10) but does not provide any reliable information regarding the quantification method. An online search of a standard database (JCPDS database) for X-ray powder diffraction patterns enables phase identification for a large variety of crystalline phases in a sample should be used and the JCPDS file to be added in the text.

4. The authors should pay attention to the figure caption in the manuscript by adding more descriptions.

5. All the experimental data (tabled results or graphical presented) is without any trace of error. Please add the error bars to the results.

Publishing these results, I believe will require a wholesale re-work of the manuscript, more than a major revision given that substantially more data is needed to support the claims.

The effort made by the authors in making this manuscript is to be appreciated, but unfortunately, as it stands, I can only propose that the paper be rejected in this form

Author Response

Point 1: Does the title, seen in isolation, give a full yet concise and specific indication of the work reported? The authors should pay attention to assuring that the title of their study is clear and informative, and should reflect the aim and approach of the work. Moreover, at a simple search, It was found the title looks similar to other already published papers. So please rewrite it with the respect to the above-mentioned characteristics. 

Response 1: Thank you very much for your comments. The title has been rewrited. Please check the revised manuscript.

Point 2: The Abstract of the present study is poorly written and lacks important information, it is, therefore, the duty of the author to ensure that the abstract is properly representative of the entire paper giving more data gathered out during the experiments.

Response 2: Thank you very much for the above comment. The Abstract has been rewrited. Please check the revised manuscript.

Point 3: The quality of figures and graphs are of very poor quality. The authors should consider that graphs, figures, and tables can save readers time and energy, aid their understanding of a scientific article, and reduce the word count of the main text. All graphs and figures as presented in the present manuscript fail to meet their potential and include mistakes that jeopardize their clarity. When the appropriate format is used to depict data, it conveys the greatest amount of information in the clearest fashion, complements the text, and deepens readers' understanding.

On the other hand, I found some of the descriptions of some very important points were inadequate or completely missing. I have very little confidence in one important analysis such X-ray diffraction, and came away with too many questions to be able to recommend this paper for publication as it stands.. I would recommend to the authors to present the result as superimposed patterns of the samples. As an exemplification, the authors provide the X-ray diffractometry results (Fig 8 and Fig 10) but does not provide any reliable information regarding the quantification method. An online search of a standard database (JCPDS database) for X-ray powder diffraction patterns enables phase identification for a large variety of crystalline phases in a sample should be used and the JCPDS file to be added in the text.

Response 3: Thank you very much for the reviewer's helpful reminder. We are very sorry that the quality of the pictures in our paper has caused trouble to the reviewers. We try to improve the clarity of the picture through software, but the effect after processing is not obvious.

The X-ray diffraction test is completed by the experimental testing center of an external unit. The X-ray diffractometry experimental results provided to us after the test are a single picture, as shown in Fig 8 and Fig 10, and we cannot modify it. In addition, the upper right corner of the data graph of the X-ray diffractometry experimental results identifies the crystalline products according to different colors.

Point 4: The authors should pay attention to the figure caption in the manuscript by adding more descriptions.

Response 4: Thank you for your suggestion. We have revised the caption of the figure. Please check the revised manuscript.

Point 5: All the experimental data (tabled results or graphical presented) is without any trace of error. Please add the error bars to the results.

Response 5: Thank you very much for the above comments. The experimental data (tabled results or graphical presented) are determined according to the average value of all parallel experimental results. However, the experiment of some test conditions was not carried out due to the damage of parallel samples. In order to ensure the beauty and uniformity of the experimental graph in the manuscript, the error bars are not marked.

Reviewer 2 Report

The experimental campaign of this study is very poor. Only one mortar studied.

The study focuses on carbonation. 

We do not see the scientific contribution. It looks like a technical report.

We do not know the type of cement, no study in the fresh state of the mortar. No information on the mixture of materials.

How to properly interpret the results if the repetition of mortars is not mastered or validated?

Is the formulation of the mortar linked to a standard?

Why a 1:1 ratio Cement sand.

Why sift the sand to 3mm?

It is not known if the analyzes are done on several samples

The study is made with a succession of techniques, we do not see the interest

Author Response

Point 1: We do not know the type of cement, no study in the fresh state of the mortar. No information on the mixture of materials.   

Response 1: Thank you very much for your comments. The cement type selected in the paper is composite Portland cement P•C 32.5R from Huaxin Cement (Wuhan) Co., Ltd. The cement mortar mixture was prepared with an effective water/cement ratio (W/C) of 0.4, and the mass ratio of cement to yellow sand was 1:1. It has been explained in Section “2.1 Materials”of the paper.

Point 2: How to properly interpret the results if the repetition of mortars is not mastered or validated?

Response 2: Thank you very much for the above comments. We carried out parallel sample tests in all test conditions. The experimental data are determined according to the average value of all parallel experimental results. Therefore, the experimental data can be interpreted the results and also used for regularity research

Point 3: Is the formulation of the mortar linked to a standard?

Response 3: Thank you very much for your comments. The formulation of the mortar is mot linked to a standard.

Point 4: Why a 1:1 ratio Cement sand.

Response 4: Thank you very much for your comments. The main purpose of this paper is to study the effects of groundwater immersion and groundwater level fluctuation on the carbonation characteristics of cement-based materials in underground engineering, which are very different from the carbonation characteristics in air. The cement-based materials in underground engineering has the characteristics of inhomogeneous, and the water environment in which it is located is relatively complex. In order to complete the qualitative research on the regularity of carbonation's influence on cement-based materials of underground engineering, cement mortar is used instead of concrete materials in this paper. Through the comparative study of mortars with different ratios, we finally choose the ratio of 1:1.

Point 5: Why sift the sand to 3mm?

Response 5: Thank you very much for your comments. The content of sand larger than 3mm is relatively small. In order to avoid the variability between samples and improve the homogeneity of the samples, sand was passed through a 3mm sieve.

Point 6: It is not known if the analyzes are done on several samples

Response 6: Thank you very much for your comments. This paper mainly analyzes the 1:1 ratio of cement and mortar. The analysis of different ratios of cement and mortar will be the focus of our follow-up research work.

Reviewer 3 Report

The manuscript entitled: “Experimental Study on Carbonation of Cement-based Materials under Water-Immersion and Saturated-Humidity Environment”, reports a study paying attention to the phenomenon of carbonation on cementitious materials. In particular, the carbonation related to a deviation tunnel was considered since its conditions alternate states of immersion in water and saturated humidity.

To study the effects of carbonation, the authors created cement-based specimens in the laboratory, subjected to conditions that simulate those to which a deviation tunnel is generally subjected.

The specimens were tested with a series of instrumental investigations.

Reading the manuscript, I found a good organization and exposition of the research. The tests were carried out with scientific rigor. The discussion of the data is in-depth and the conclusions are consistent with the reported results.

I have no particular suggestions to send to the authors.

In light of the above, I believe that the work can be considered for its publication.

Author Response

Point 1: I have no particular suggestions to send to the authors. In light of the above, I believe that the work can be considered for its publication.

Response 1: Thank you for your support and encouragement.

Round 2

Reviewer 2 Report

Dear, I accept the corrections made. Regards